# The Effects of Aging on the Intramuscular Connective Tissue

**DOI:** 10.3390/ijms231911061

**Published:** 2022-09-21

**Authors:** Caterina Fede, Chenglei Fan, Carmelo Pirri, Lucia Petrelli, Carlo Biz, Andrea Porzionato, Veronica Macchi, Raffaele De Caro, Carla Stecco

**Affiliations:** 1Department of Neurosciences, Institute of Human Anatomy, University of Padua, 35121 Padua, Italy; 2Department of Surgery, Oncology and Gastroenterology, Orthopedic Clinic, University of Padua, 35128 Padua, Italy

**Keywords:** aging, intramuscular connective tissue, extracellular matrix, muscle, collagen, hyaluronan, motor control

## Abstract

The intramuscular connective tissue plays a critical role in maintaining the structural integrity of the muscle and in providing mechanical support. The current study investigates age-related changes that may contribute to passive stiffness and functional impairment of skeletal muscles. Variations in the extracellular matrix in human quadriceps femoris muscles in 10 young men, 12 elderly males and 16 elderly females, and in the hindlimb muscles of 6 week old, 8 month old and 2 year old C57BL/6J male mice, were evaluated. Picrosirius red, Alcian blue and Weigert Van Gieson stainings were performed to evaluate collagen, glycosamynoglycans and elastic fibers. Immunohistochemistry analyses were carried out to assess collagen I, collagen III and hyaluronan. The percentage area of collagen was significantly higher with aging (*p* < 0.01 in humans, *p* < 0.001 in mice), mainly due to an increase in collagen I, with no differences in collagen III (*p* > 0.05). The percentage area of elastic fibers in the perimysium was significantly lower (*p* < 0.01) in elderly men, together with a significant decrease in hyaluronan content both in humans and in mice. No significant differences were detected according to gender. The accumulation of collagen I and the lower levels of hyaluronan and elastic fibers with aging could cause a stiffening of the muscles and a reduction of their adaptability.

## 1. Introduction

The age-related decline of locomotor ability affecting muscle strength, proprioception, and motor control is a well-established phenomenon [1,2]. Several studies have demonstrated that these age-related alterations could be caused by the physiological loss of skeletal muscle mass (sarcopenia) leading to the onset of atrophy [3], which leads to decline in functional performance and reduced flexibility, coupled with a decrease in strength and force output. The age-related sarcopenia is caused by the progressive incapacity of the regenerative machinery to replace damaged muscles [4]. The consequent muscle weakness and postural instability are major contributors to the incidence of falls in the elderly [5].

Skeletal muscle is made up of muscle fibers, which are embedded in the intramuscular connective tissue (IMCT), which, in turn, is made up of the epimysium, perimysium and the endomysium [6] and is part of a continuous network of tendons, periosteum, aponeurotic fasciae, ligaments and joint capsules, interlinking with various structures throughout the body’s musculoskeletal system [7]. The IMCT plays a key role in force transmission and motor coordination and it contributes to the elastic tissue response of the muscle [8]. Furthermore, the IMCT is a biological reservoir of muscle-resident stem cells [9]: it is recognized that striated muscle is highly plastic and capable of regeneration. During growth, all components of the musculoskeletal system (muscles, tendons, connective tissue, nerves) coordinate their growth and differentiation [10]. In the muscle’s connective tissue, some cells with mesenchymal stem properties act as fibro-adipogenic progenitors: they work as cellular sentinels for adult muscle homeostasis and regeneration, secreting extracellular matrix components, cytokines and immune-modulatory factors, and act as precursors of fibroblasts, adipocytes and osteogenic cells [11]. In addition, the blood vessels and nerves travel together within the IMCT following their 3-D organization [12]. This dynamic environment permits the homeostasis of the muscle, giving rise to a functional unit capable of executing elaborate movement.

From a microscopic viewpoint, the IMCT consists of two main components: cells and the extracellular matrix (ECM), which is essentially made up of protein fibers (collagen and elastic fibers) and ground substance (proteoglycans, multi adhesive glycoproteins, glycominoglycans and water). The resident cells, such as fibroblasts, produce collagen and elastic fibers and maintain the ECM, while the transient/wandering cells play an important role in the body’s defense system. One of the key elements of glycosaminoglycans is hyaluronan (HA), a mega-dalton glycosaminoglycan polymer critical for the integrity of the ECM and for the action of retaining water [13].

The ECM plays a key role in the structural characteristics of the skeletal muscle as it provides a three-dimensional scaffolding for muscle fibers [6]. It regulates muscle development and it is essential for muscle contraction and force transmission, regulating at the same time cell proliferation, adhesion, migration, polarity, differentiation and apoptosis [14]. Recent biomechanical studies have shown that the ECM bears most of the muscle’s passive load, which means that an individual’s range of movement and stiffness mainly reflect the ECM characteristics [15,16]. Muscle pathology is usually described in terms of altered fiber type or size, and the importance of the ECM on the mechanical transmission of force is quite an open debate. However, some authors reported that pathological changes in skeletal muscle are associated with ECM fibrosis [17]. Theret and co-authors (2021) highlighted that changes in matrix quantity and quality, and an increased matrix and tissue stiffness may help tip the affected tissue into a self-perpetuating pathological state [18]. The role of alterations of the ECM between muscle fibers in skeletal muscle stiffness was already recently demonstrated and linked to a collagen accumulation [19]. Furthermore, some recent studies affirmed that aging can improve the differentiation of fibro-adipogenic progenitors to a fibrogenic state [17,20].

This study aims to shed more light on age-related modifications in the ECM of skeletal muscles to try to add new information on the aging mechanisms of muscles and to the occurrence of aging-associated stiffness and functional impairment.

## 2. Results

### 2.1. The Human Participants

The characteristics of the 38 subjects included in this study are shown in Table 1: the men subjects had no statistically significant differences in the height, weight or body mass index (BMI) (young vs. elderly, *p* > 0.05). The groups of elderly men and elderly women showed no differences in age and BMI (*p* > 0.05), but significant differences for height and weight (*p* < 0.01).

### 2.2. The Collagen Contents and Elastic Fibers of the ECM in the IMCT

Muscle atrophy (Figure 1B) and myosteatosis (Figure 1A) were present in no less than 75% of the elderly human samples (Table 2), especially in the subjects over 70 years of age.

The percentages of collagen fibers in the IMCT (endomysium plus perimysium) were: 5.99 ± 1.34% in the young men, 10.02 ± 3.69% in the elderly men, and 9.90 ± 1.50% in the elderly women (Table 2). For the endomysium alone, the percentages were: 2.95 ± 1.59% in the young men, 6.87 ± 1.79% in the elderly men, and 6.58 ± 1.12% in the elderly women. The collagen contents in the IMCT were significantly elevated in the elderly subjects (*p* = 0.001) (Figure 1C–G; Figure 2; Table 2), with no significant differences according to gender (*p* > 0.05) (Table 2).

WVG staining (Figure 3A–D) and TEM images (Figure 3E,F) detected elastic fibers in the endomysium and perimysium. The area percentage of elastic fibers in the perimysium was significantly lower (*p* = 0.001) in the elderly men group (3.50 ± 1.66%) with respect to that in the young men group (7.99 ± 1.65%) (Table 3, Figure 3G).

Immunohistochemical staining (Figure 4) showed that COLI and COLIII were localized in the endomysium and perimysium. COLI in the elderly men’s group (AOD 0.37 ± 0.04) was significantly higher compared to that found in their younger counterparts (AOD 0.25 ± 0.04; *p* = 0.001), whereas the differences in COLIII were not significant (Young men: AOD 0.33 ± 0.09; Elderly men: AOD 0.28 ± 0.07; *p* = 0.293) (Table 3, Figure 4E,F).

Moreover, in the mice, the COLI and COLIII distribution was evident in the perimysium and endomysium (Figure 5). The area percentage of collagen fibers changed according to the age: it was equal to 2.05 ± 0.44% in Group A, 3.26 ± 0.86% in Group B, 7.97 ± 0.80% in Group C, showing that the collagen content significantly increased in the older mice (Group A vs. Group B: *p* = 0.071; Group B vs. Group C: *p* < 0.001; Group A vs. Group C: *p* < 0.001) (Table 4, Figure 6A,B). The accumulation of COLI significantly increased with aging (Group A: AOD 0.17 ± 0.01; Group B: AOD 0.22 ± 0.02; Group C: AOD 0.30 ± 0.03; Group A vs. Group B: *p* = 0.013; Group B vs. Group C: *p* = 0.002; Group A vs. Group C: *p* < 0.001; Table 4), but the differences in COLIII were not significant (Group A: AOD 0.23 ± 0.08; Group B: AOD 0.23 ± 0.04; Group C: AOD 0.22 ± 0.05; *p* > 0.05; Table 4).

Picrosirius red staining (Figure 6) confirmed the distribution of collagen fibers both in the endomysium and perimysium of human (Figure 6C,D) and mouse (Figure 6A,B) specimens.

The color distribution under polarized light showed changes in the organization and density of collagen fibers according to age (Figure 6E,F): changes in color (shifts from orange/red to yellow/green) are believed to show changes in the organization, configuration and direction of the collagen fibers [21].

### 2.3. HA in the ECM

Alcian blue staining showed that the IMCT was rich in glycosaminoglycans (Figure 7A, Alcian blue 0.05% with MgCl_2_ 0.05 M). Furthermore, the decreased staining associated with the increased electrolyte concentration (Figure 7B, Alcian blue 0.05% with MgCl_2_ 0.3 M) lends support to the notion that the staining is due to a HA-rich matrix, as the HA selectively stained only a low concentration of salt (data obtained both in young and elderly, not shown) [22]. Biotinylated HABP immunohistochemistry staining showed that HA was localized in the endomysium and perimysium in the human samples (Figure 7C,D). The HA content was significantly decreased in the elderly men with respect to the content in their younger counterparts (Elderly men: 3.16 ± 2.18 µg/g; Young men: 6.83 ± 2.91 µg/g, *p* = 0.04, Figure 7E), but the difference between the elderly men and women groups was not significant (Elderly women: 3.87 ± 2.44 µg/g; *p* > 0.05, Figure 7F).

The same results were obtained in the mice: Alcian blue (data not shown) and immunohistochemistry showed HA in the epimysium, perimysium and the endomysium (Figure 7G–I). HA significantly decreased with aging (Group A: 16.84 ± 5.32 µg/g; Group B: 9.33 ± 1.16 µg/g, *p* < 0.05; Group A vs. Group C: 7.32 ± 2.21 µg/g, *p* < 0.01, Figure 7J).

## 3. Discussion

### 3.1. Collagen and Elastic Fibers in the IMCT of Skeletal Muscle with Aging

The collagen levels were significantly higher in the ECM in both the elderly human and mouse groups with respect to their younger counterparts; this was mainly due to COLI. No significant differences were found in the COLIII in neither the humans nor the mice, and no differences were detected according to gender.

Our results are in line with some histological and biomechanical studies investigating various animal species [23,24], although Haus et al. reported that endomysial collagen was unchanged in elderly adults [25]. One study by Karsdal et al. (2016) demonstrated that the COLI turnover was downregulated by up to 93% in one year old rats compared to one month old rats [26]. In general, COLI provides resistance to force, tension and stretch, while COLIII provides a flexible meshwork for cellular support and a supportive scaffolding for muscle fibers [27,28]. Age-related alterations in the collagen content in the ECM with a considerable increase of COLI in elderly muscle may therefore reduce the autonomous gliding and deformable property of the endomysium during muscular fiber contraction and affect the motor coordination and force transmission linked to the perimysium.

Little information is currently available on the amount of elastic fibers in the ECM. According to one study examining aging in rats (1, 4, 8 and 18 month old rats) the density of mature elastic fibers was found to be progressively higher with age, while the amount of resistant oxytalan fibers decreased in both the diaphragm and the rectus abdominis [29]. Another study showed that the elastin turnover was not significantly influenced by age in rats [26]. It should, nevertheless, be noted that the oldest rat in those studies corresponded to less than 45 years of age for humans. In our study, the percentage area of elastic fibers in the perimysium was significantly lower in the elderly men group compared to that in their younger counterparts. The finding agrees with research by Stearns et al. (2017) who reported that the percentage area of the elastic fibers in the ECM was significantly lower in the elderly mice (22–24 month old) with respect to their younger counterparts (3–4 month old) and confirmed the hypothesis of an age-related increase in the stiffness in skeletal muscle [17]. The elastic fibers can stretch up to 1.5 times their length and snap back to their original length when relaxed [30]. Elastic fibers in the ECM are interwoven with collagen fibers to limit the distensibility of the tissue and to prevent tearing from excessive stretching. Age-related alterations in the elastic fiber component could reduce the tissue elasticity and compromise the elastic fiber function and force transmission.

### 3.2. HA in the IMCT of the Skeletal Muscle with Aging

HA constitutes space filling, it plays a significant role in muscle repair and regeneration in the event of muscle injury and it promotes the migration and proliferation of myogenic cells by interacting with the CD44 and RHAMM receptors [31]. Migration, proliferation, differentiation and fusion of the mononuclear mycoplasmas to the multinucleated myofibers are mediated by the IMCT surrounding them [32]. From a biomechanical point of view, under normal conditions, large HA polymers can absorb large volumes of water, making the polymer an excellent lubricant and shock absorber [27] and allowing collagen fibers to slide with less friction during movement [33].

In this study we showed that the HA content was significantly lower in the IMCT of elderly human subjects and mice studied, as already demonstrated in the skin as a physiological process influenced by aging [34,35,36]. This HA reduction with aging, along with the accumulation of collagen content and the reduction in elastic fibers inhibits the tissue from absorbing water and contributes to become much more rigid and less elastic, with altered gliding properties.

### 3.3. The IMCT: A Clinical Viewpoint

The age-related alterations of the ECM embedded in the IMCT demonstrated by this study, can be associated with alterations of the muscle properties. In particular, the increase in COLI together with a decrease in HA and elastic fibers can have an impact on muscle mechanics, coordination and lubrication, and can contribute to reduce the muscles’ force transmission and general flexibility, joint mobility and locomotor activity [37]. The high range of motion (ROM) and motor abilities in children can be due to the richness in HA of the IMCT and to the collagen fibers, not completely spatially organized, and consequently very adaptable in multiple directions [38].

We recently demonstrated that the fasciae during the fetal development are similar to “blank sheets” composed of few elastic fibers, abundant collagens and HA, on which various forces, fetal movements, loads and gravity, act to organize the tissue and the collagen fiber orientation [39]. At the same way, after birth and during life, the movements can structure the fascial system. During the skeletal muscle development process and motor learning that take place between puberty and adulthood, the ECM becomes more organized and better defined by acquiring a specific peripheral spatial organization with the input of mechanical stimuli (such as movement patterns), and hormones. During puberty, the hypothalamic-pituitary-gonadal axis activity peaks, the alterations occur in the circulating gonadotropin levels with elevated levels of sex steroids. One of our previous studies confirmed that fibroblasts are sensitive to sex hormones, which can modulate the production of some components of the ECM depending on hormone levels [40]. Doessing et al. (2005) suggested that the growth hormone promotes the matrix collagen synthesis in muscolo-tendinous tissue [41]. According to this evidence, it is likely that changes in hormone levels (growth hormone, sex steroids) that take place during puberty can stimulate the receptors of fibroblasts in the IMCT [42,43], with a further regulation of the ECM components to meet the needs of skeletal muscle growth and development [44].

With aging, a decrease of HA occurs together with a increase of collagen, creating a more rigid IMCT. In addition, inactivity and sedentary behaviors in the elderly cause a decline in skeletal muscle metabolism, together with a reduction of the ability of motor learning, range of motion and new locomotor patterns [45]. The same age-related process was demonstrated in the ECM of muscle spindles in mice samples: the increased stiffness partly explains the peripheral mechanisms underlying the age-related decline of proprioception and motor coordination [46].

This work confirmed the age-related modifications of the ECM of the IMCT, therefore confirming the ECM’s primary role in muscle functions. Other animal and human biomechanical studies demonstrated that skeletal muscle stiffness with aging is due to alterations of the ECM rather than of the muscle fibers [17,19,47].

Thus, in our opinion, it will be necessary, in the near future, to evaluate whether specific training regimens and non-sedentary behaviors can slow down this IMCT evolution in elderly subjects, thus permitting a better maintenance of the passive muscle elasticity and flexibility [48,49,50].

### 3.4. Limitations and Further Research

Several limitations of this study should be declared. Firstly, no samples of skeletal muscle of young female or children were available for this study, thereby preventing a better exploration of the role of sex hormone levels and gender in the contents of the ECM. Our analysis of human specimens was based on only a small population sample: further studies with larger sample sizes and subjects that are more diverse are needed to establish the reliability and validity of our findings. Furthermore, the amount of HA compared to the other GAGs has not been explored in order to better understand the relative modifications with aging. Moreover, the function of HA is not only related to its concentration but also to its molecular weight, which remained unexplored in this study. In fact, studies have confirmed that high molecular mass HA is considered a physiological protector of cells, acting as a scaffold for proteoglycans, whereas low molecular mass HA fragments can stimulate pro-inflammatory cellular responses [51]. Lastly, in this work we have no data about the orientation, crimping and thickness of the collagen fibers at the ultrastructural level: for this reason, in the near future, it would be necessary to perform a second harmonic generation analysis, together with an immunogold labeling of individual collagen fibers, to achieve more precise information about the distribution of collagen type I and III.

## 4. Materials and Methods

### 4.1. Collection of Human Specimens

Thirty-eight patients who underwent surgery for a femur fracture at the Padova Orthopaedic Clinic between January 2018 and July 2020 were considered eligible for the study. The study design was approved by the institution’s Ethics Committee (Studio 3027P/AO/13), and the patients were fully informed about the modality and aims of the study before they were asked to sign consent statements.

Subjects with medical conditions that could cause peripheral neuropathy, such as diabetes mellitus, malignancy and other endocrine diseases, were excluded from the study. The patients were classified into groups depending on their age and sex: 10 young men (<50 years, average age: 37.5 ± 9.5 years), 12 elderly men (>50 years, average age: 79.0 ± 12.4 years), and 16 elderly women (>50 years, average age: 80.6 ± 11.4 years). No young females (<50 years) or children were eligible to participate.

The samples of the quadriceps femoris (23 vastus lateralis and 15 rectus femoris) muscle were collected: half of these were immediately post-fixed in 10% neutral buffered formalin (10% NBF), pH 7.4, for histological and immunohistochemistry analyses, and the others were snap frozen in liquid nitrogen and stored at −80 °C before the determination of the HA level.

### 4.2. Collection of Animal Samples

Fifteen male C57Bl/6J mice (five 6 weeks old = Group A, five 8 months old = Group B, and five 2 years old = Group C) were provided by the University of Padova’s Animal Center (Padova, Italy). All animal procedures were approved by the ethical committee of the University of Padova, in agreement with the guidelines of the Italian Department of Health, and in compliance with the ARRIVE (Animal Research: Reporting of In Vivo Experiments) guidelines.

The ages of the mice corresponded approximately to 11.5, 35 and 70 in human years, respectively [52]. The mice were kept in cages in an environmentally controlled room with the temperature adjusted to 22 °C in which there were diurnal light-dark cycles and free access to water and food.

Once the hindlimb skin was removed, the entire left hindlimb including the tibio-fibula bone was immediately post-fixed in 10% NBF, pH 7.4, before decalcification and the histological and immunohistochemistry studies. In this case, the tibio-fibula bone was preserved intact to maintain the overall morphology and interrelationship of the muscles for further specimen processing, avoiding any damage of the IMCT within the muscles. The right tibio-fibula bone of the animals was instead removed to collect the hindlimb muscles (including the gastrocnemius, soleus, tibialis anterior and posterior, fibularis longus and brevis), and then quickly frozen in liquid nitrogen and stored at −80 °C to quantify HA.

### 4.3. Decalcification Protocol for the Mouse Hindlimbs

In this study, the decalcification was performed using 10% ethylenediaminetetraacetic acid (EDTA): using this method, the decalcification rate was slow, but the tissue’s antigenicity was preserved for the immunohistochemical staining. The protocol used was as follows: the fixed hindlimbs were placed in PBS (20 min × 3) and distilled water (20 min × 3). Then, the hindlimbs were placed in at least 15 volumes of 10% EDTA which was changed weekly, over a 2 week period. Finally, the hindlimbs were rinsed in distilled water (20 min × 2) for paraffin embedding, sectioning and histological and immunohistochemistry staining.

### 4.4. Histological Stainings and Morphological Analysis of the IMCT

Once the samples were paraffin-embedded, 5 μm cross-sections were cut and stained with hematoxylin and eosin (H&E) to examine the tissue morphology, Picrosirius red staining and to identify the collagen content and the Weigert Van Gieson (WVG) staining to evaluate the elastic fibers (black-violet in color). In addition, slides were incubated in Alcian blue solutions (0.05%, pH 5.8) with different MgCl_2_ concentrations (0.05 M; 0.3 M; 2 M) to stain the acidic polysaccharides and glycosaminoglycans (blue in color). As described in the literature [22,53], different concentrations of MgCl_2_ cause differences in the staining: the cations of the salt compete with those of the dye for the polyanionic sites in the tissue, permitting a selective stain of specific glycosamynoglycans.

### 4.5. Transmission Electron Microscopy (TEM) and Ultrastructural Analysis of the IMCT

Two specimens from each of the human male groups (young and elderly) were randomly selected and fixed in 2.5% glutaraldehyde (Serva Electrophoresis, Heidelberg, Germany) in 0.1 M phosphate-buffered, post-fixed in 1% osmium tetroxide (OsO4) (Agar Scientific Elektron Technology, Stansted, UK) in 0.1 M phosphate buffer, dehydrated in a graded ethanol series, and embedded in Epoxy Embedding Medium Kit (45349, Sigma-Aldrich, St. Gallen, Switzerland). Ultrathin (60 nm) and semi-thin (0.5 µm) sections were cut with a RMC PowerTome ultramicrotome (Boeckeler Instruments, Tucson, AZ, USA) and stained with 1% toluidine blue at 80 °C. The ultrathin sections were collected on 300-mesh copper grids and counterstained with 1% uranyl acetate and then with Sato’s lead solution. The specimens were examined using a Hitachi H-300 Transmission Electron Microscope.

### 4.6. Immunohistochemistry Staining: Analysis of Collagen Type I (COLI), Collagen Type III (COLIII) and Hyaluronic Acid Binding Protein (HABP)

Five-micron-thick sections were cut and treated with a solution of H_2_O_2_ 1% for 15 min to inhibit any endogenous peroxidase activity. The slides for COLI and COLIII were treated using a heat-induced antigen retrieval with citrate buffer pH 6 in 80 °C for 20 min, followed by three washings in PBS. Following the treatment with 0.1% bovine serum albumin (BSA) for 1 h, all of the slides were incubated with primary antibodies: Goat Anti-Collagen I (1:400, Southernbiotech), Rabbit polyclonal anti-Collagen III (1:400, ab7778 Sigma), biotinylated HABP (1:900 Millipore) in BSA at 4 °C overnight. Following three washings in PBS, the sections were incubated with the secondary antibodies: anti-rabbit IgG peroxidase-conjugated antibody for COLIII (1:200), anti-Goat peroxidase-conjugated antibody for COLI (1:300), HRP conjugated Streptavidin for HABP (1:250) (Jackson ImmunoResearch, Cambridgeshire, UK) for 1 h. Following repeated washings in PBS, the reaction was developed with 3,3′-diaminobenzidine (Liquid DAB substrate Chromogen System kit Dako Corp, Carpinteria, CA, USA). The negative controls were obtained by omitting the primary antibodies. Each slide was counterstained with hematoxylin (Dako Corp.) and then dehydrated in a graded ethanol series and mounted for microscopic evaluation.

### 4.7. Purple-Jelley HA Assay: Quantification of Hyaluronan in the IMCT

The Purple-Jelley HA assay (Biocolor Ltd.) was used to measure the HA content in human and mouse skeletal muscles [53]. Briefly, 250 mg ± 50 mg human and mouse muscle tissues were cut into small fragments with a surgical scalpel and then transferred to 2.0 mL microcentrifuge tubes and digested at 55 °C in 400 µL TRIS–HCl (50 mM, pH 7.6) containing Proteinase K (Sigma) overnight. Following the centrifugation at 12,000 rpm for 10 min, the supernatants were mixed with 1.0 mL a GAG precipitation reagent in new microcentrifuge tubes. Following the centrifugation at 12,000 rpm for 10 min, the resulting residues were mixed in NaCl and cetylpyridinium chloride (CPC) in water. Once these steps were repeated and the total GAG content was recovered, HA was isolated by adding 500 µL 98% ethanol, then centrifuged and fully hydrated in 100 µL of water. For the colorimetric analysis, 200 µL of a purple dye reagent was added to 20 µL aliquots of the test samples or standards or reagent blanks. The absorbance value at 655 nm was read using a Tecan Infinite M1000 Pro microplate reader (San Jose, CA, USA) in 96-microwell plates (twice for each sample). The absorbance values of the standard curve obtained with the HA standard (200 µg/mL), was converted into µg HA contained in the total volume of 100 µL. Finally, after the calculation of the µg HA extracted from the starting tissues samples, the average µg HA per gram of wet muscle tissue was obtained for each sample (mean of at least two measurements ± standard deviation). A reference was made to the results obtained previously from four adult human skin samples as control data to validate the extraction method [53].

### 4.8. Image Analysis

The specimens were photographed using a Leica DMR microscope (Leica Microsystems, Wetzlar, Germany) and the images were analyzed with ImageJ software, freely available at http://rsb.info.nih.gov/ij/ (accessed on 1 September 2018).

The samples were stained with Picrosirius red were analyzed at 20× and 10× enlargement for the human samples, and 40× for the mouse specimens. The field area containing perimysium (including the primary, secondary and tertiary fascicles) and endomysium was used to estimate the collagen amount. At least 20 fields from five sections of each specimen were counted in order to obtain the percentage of the collagen content (area %) of the perimysium and/or endomysium.

The elastic fibers stained by WVG were difficult to quantify in the endomysium, so the analysis of elastic fibers (area %) was performed in the perimysium at a final magnification of 40× (at least 20 images from five sections of each specimen).

To measure COLI and COLIII with immunohistochemistry staining, the readings were obtained using blanked fields at a final magnification of 10× for the human specimens and 40× for the mouse specimens containing the perimysium and endomysium. At least 20 images from five sections of each specimen were counted and the readings of the average optical density (AOD) were averaged to obtain the representative values of COLI and COLIII.

### 4.9. Statistical Analysis

All of the data management and statistical analyses were performed using IBM SPSS version 25.0 software. The Shapiro–Wilks test and Levene’s tests were performed to investigate data distribution and the homogeneity of the variance test, respectively.

The age, height, weight, body mass index (BMI) of the human participants, the percentage area of collagen in the humans and mice, the percentage area of the elastic fibers in the perimysium in humans, the AOD of COLI and COLIII and the HA content in the human and mouse sections were reported as means ± standard deviations (M ± SD), as they had a normal distribution. The differences in these data were analyzed using an unpaired two sample *t*-test to compare the results of the young and the elderly male groups and those of the elderly men and women groups. Analyses of variance (ANOVA) with the Tukey post-hoc test (normally distributed data and equality of variances assumed) or the Games–Howell post-hoc test (normally distributed data but equality of variances not assumed) were used to compare the area percentage of collagen, the AOD of COLI, COLIII, and the HA content in the mice. A *p* value ≤ 0.05 was considered the study’s limit for statistical significance.

## 5. Conclusions

The age-related variations in the ECM contents seem to affect the cellular adaptation of the IMCT resulting in alterations of the mechanical properties of the skeletal muscle. The age-related accumulation of collagen content in the ECM, the reduced number of elastic fibers, and the lower HA concentrations identified by our analysis could reduce tissue adaptability, alter the normal tissue gliding mechanisms of the IMCT and affect skeletal muscle function and contraction. These facts at least partially explain the peripheral mechanisms of age-related locomotor ability deficits.

## Figures and Tables

**Figure 1 ijms-23-11061-f001:**
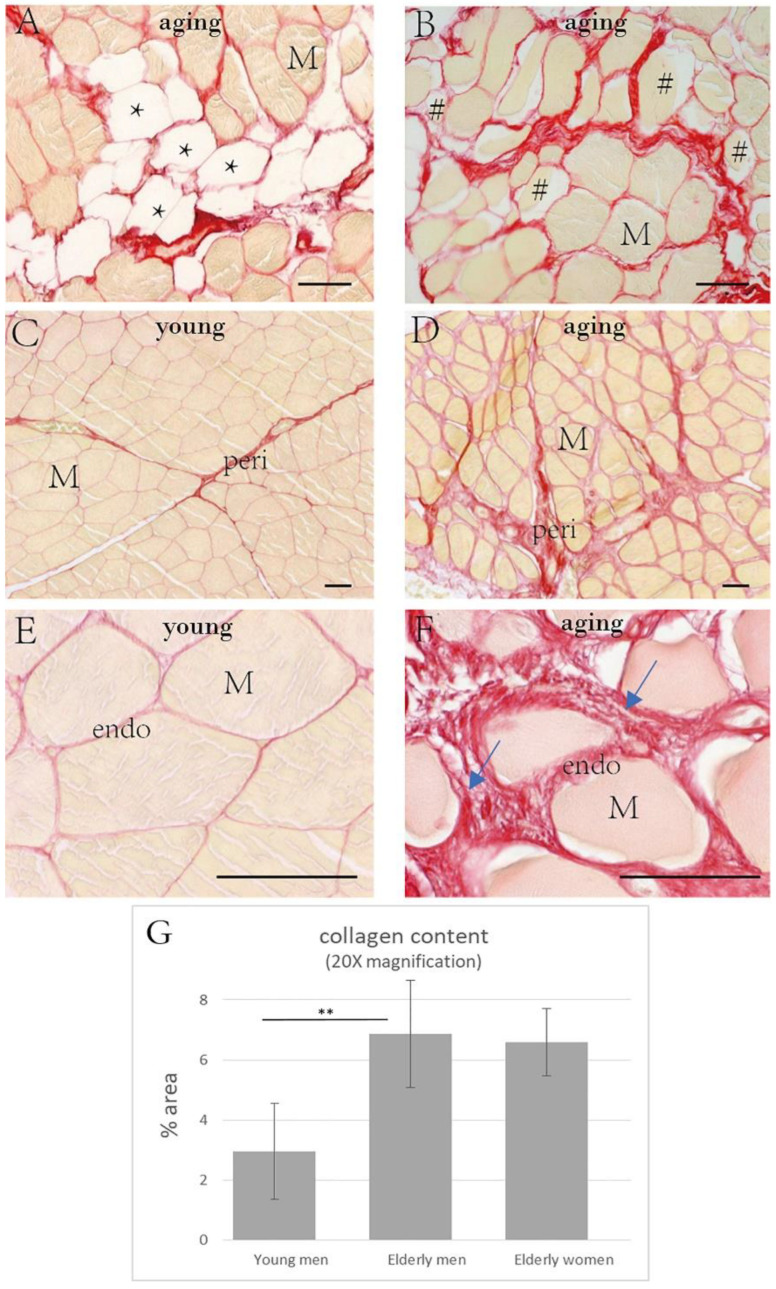
Picrosirius red staining in the human muscle samples. (**A**): muscle sample from an 87 year old man with fat infiltration (*); (**B**): muscle sample from a 94 year old woman with muscle atrophy (^#^). The collagen intensity and the disposition of the fibers stained (in red) are shown in young subjects ((**C**), 18 year old male; (**E**), 32 year old male) and in elderly men ((**D**), 83 years old; (**F**), 94 year old male). The altered arrangement of the collagen fibers can be noted in the older subject (arrows). The analysis of the collagen content (**G**) showed the higher amount of collagen in elderly men with respect to young men (** *p* < 0.01). M = muscle cell, peri: perimysium, endo: endomysium. Scale bar: 50 μm.

**Figure 2 ijms-23-11061-f002:**
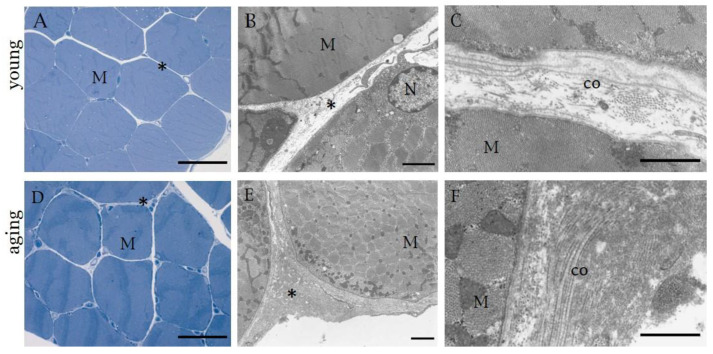
Semithin sections stained with 1% Toluidine blue (**A**,**D**) and TEM images (**B**,**C**,**E**,**F**) of human muscle samples. (**A**–**C**): 27 year old male’s muscle; (**D**–**F**): 74 year old male’s muscle. * indicates collagen of IMCT. M = muscle cell, co = Collagen fibers, N = Nucleus. (**A**,**D**): Scale bar: 50 μm; (**B**,**E**): Scale bar: 2 µm, (**C**,**F**): Scale bar: 1 µm.

**Figure 3 ijms-23-11061-f003:**
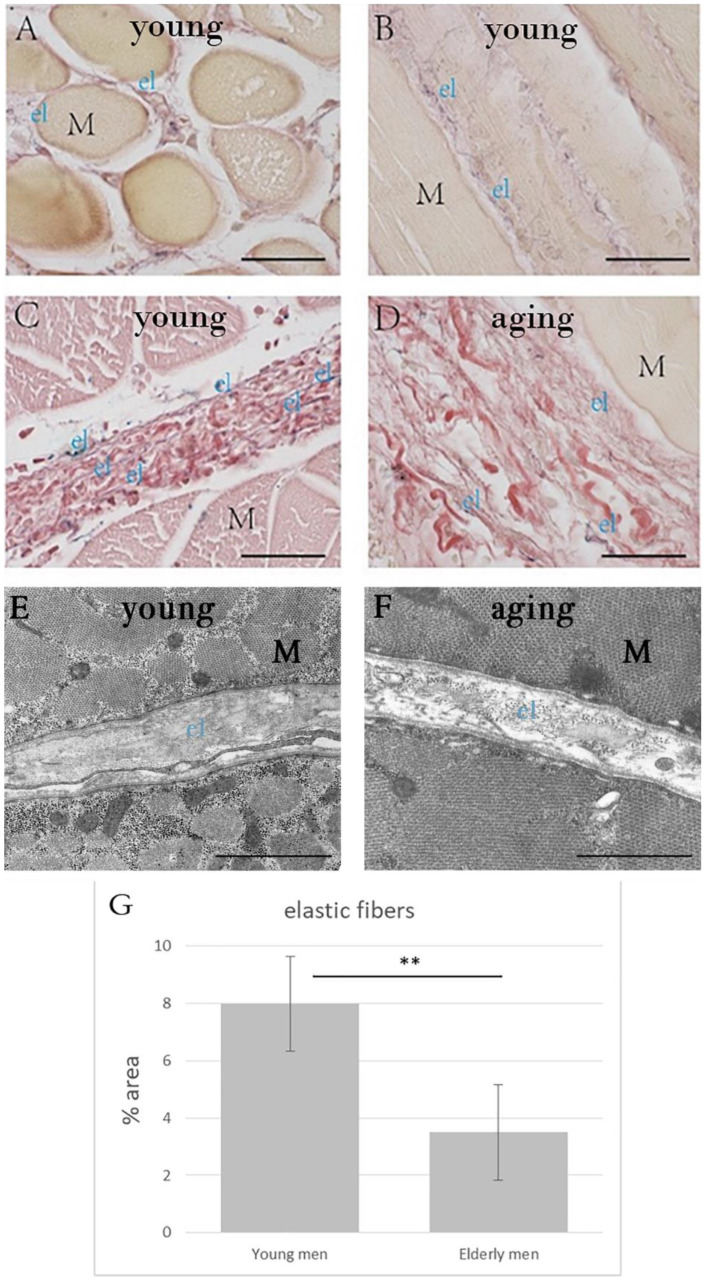
Weigert Van Gieson staining (**A**–**D**) of human muscle samples, from a 49 year old man (**A**,**B**), 18 year old man (**C**), 94 year old man (**D**). (**E**,**F**) show TEM images from 27 year old man (**E**) and 74 year old man (**F**). The amount of elastic fibers was significantly higher in young men with respect to elderly men (**G**) (** *p* = 0.01). M = muscle cell; el: elastic fibers (black-violet color) in the endomysium (**A**,**B**) and perimysium (**C**,**D**). (**A**–**D**): Scale bar: 50 μm. (**E**,**F**): Scale bar: 1 µm.

**Figure 4 ijms-23-11061-f004:**
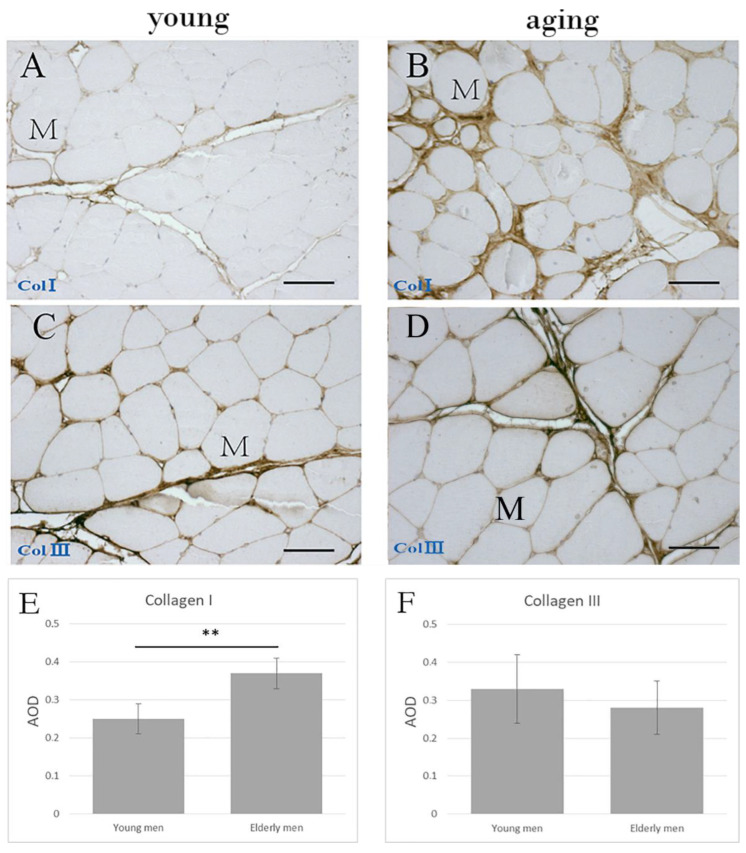
Immunohistochemical staining of collagen type I (**A**,**C**) and collagen type III (**B**,**D**) in the perimysium and endomysium of human muscle samples from an 18 year old (**A**,**C**) and an 83 year old (**B**,**D**) human patient. (**E**,**F**) show the AOD of the collagen I and collagen III, respectively, in young and elderly men. ** *p* < 0.001. M = muscle cell. Scale bars: 50 μm.

**Figure 5 ijms-23-11061-f005:**
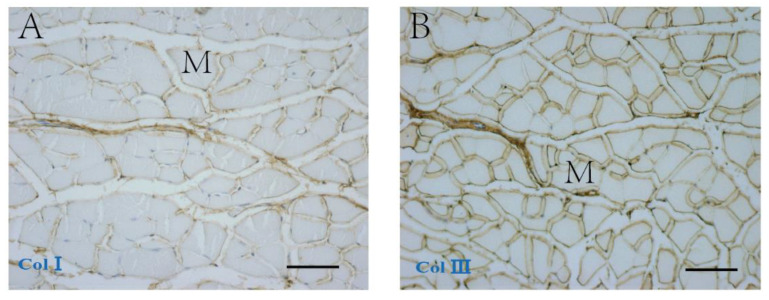
Immunohistochemical staining of collagen type I (**A**) and collagen type III (**B**) in the perimysium and endomysium of 8 month old mouse muscle (Group B). M = muscle cell. Scale bar: 50 μm.

**Figure 6 ijms-23-11061-f006:**
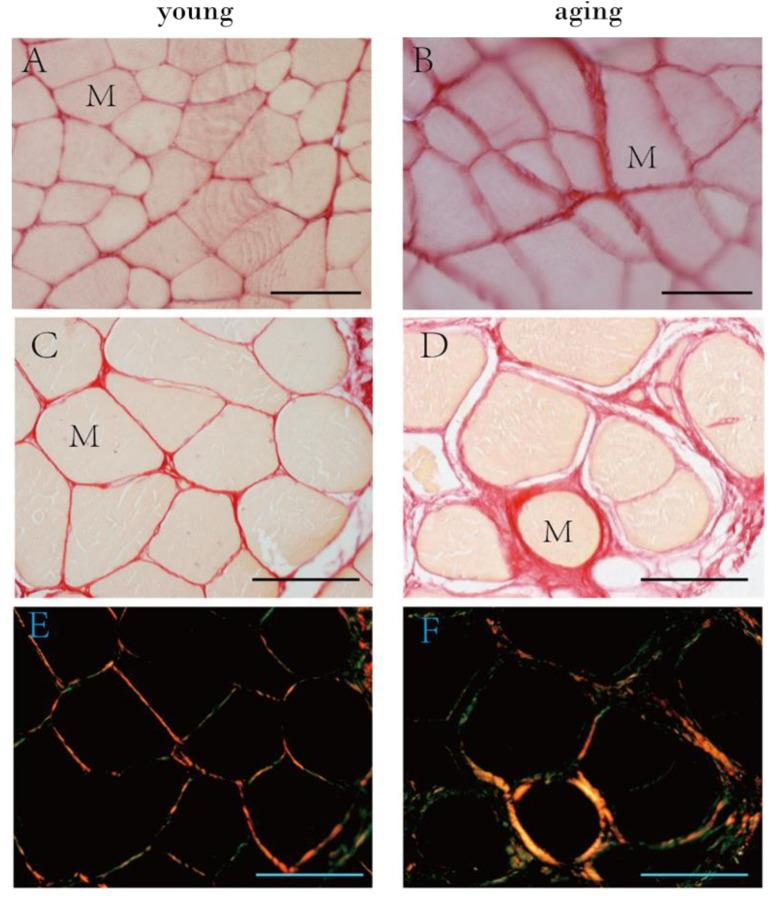
Picrosirius red staining of mouse (**A**,**B**) and human muscle samples (**C**–**F**) and polarized light images in human samples (**E**,**F**). (**A**): muscle sample from 6 week old mice (puberty, Group A), (**B**): from 2 year old mice (elderly, Group C); (**C**,**E**): muscle sample from a 32 year old human patient; (**D**,**F**): muscle sample from an 87 year old human patient. M = muscle cell. Scale bar: 50 μm.

**Figure 7 ijms-23-11061-f007:**
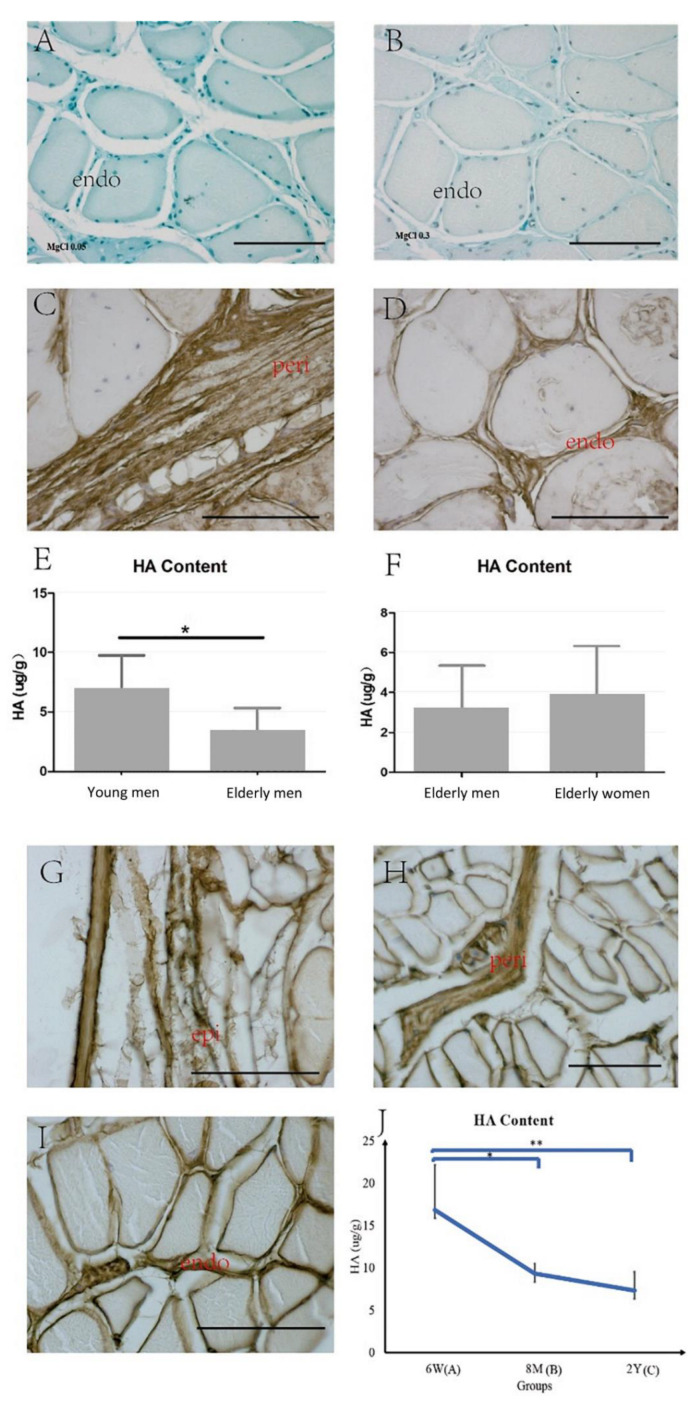
Alcian blue 0.05% with MgCl_2_ 0.05 M (**A**) and MgCl_2_ 0.3 M (**B**) of human muscle samples, and biotinylated HABP immunohistochemistry staining of the human (**C**,**D**) and mouse (**G**–**I**) muscle samples. epi: epimysium, peri: perimysium, endo: endomysium. Scale bars: 50 μm. The Purple–Jelley HA assay showed the HA amount in human ((**E**): young vs. elderly women, * *p* = 0.04; (**F**): elderly men vs. elderly women), and in the mouse skeletal muscle ((**J**), * *p* < 0.05, ** *p* < 0.01). 6 W: 6 weeks old (Group A, puberty); 8 M: 8 months old (Group B, middle age); 2 Y: 2 years old (Group C, elderly).

**Table 1 ijms-23-11061-t001:** Characteristics of the human participants (n = 38).

Characteristic	Y Men (n = 10)	E Men (n = 12)	E Women (n = 16)	Y vs. E Men *p*-Value	E Men vs. Women *p*-Value
Age (y)	37.5 ±9.5	79.0 ±12.4	80.56 ± 11.4	<0.001 ***	0.931
Sex	M	M	F	NA	NA
Muscle	6VL, 4RF	8VL, 4RF	9VL, 7RF	NA	NA
Height (cm)	177.0 ± 6.4	176.8 ± 5.0	165.1 ± 3.7	0.919	<0.001 ***
Weight (kg)	75.0 ± 6.9	77.3 ± 9.3	63.9 ± 9.6	0.519	0.001 ***
BMI (kg/m^2^)	23.9 ± 1.5	24.8 ± 3.1	23.5 ± 4.0	0.429	0.624

Unpaired, two-sample *t*-test, NA: not applicable, Values are presented as numbers or means ± SD. VL: vastus lateralis, RF: rectus femoris. BMI: body mass index, Y: Young; E: Elderly. *** *p* ≤ 0.001.

**Table 2 ijms-23-11061-t002:** The collagen component in the ECM of the IMCT of the human skeletal muscle (n = 38).

Characteristic	Y Men (n = 10)	E Men (n = 12)	E Women (n = 16)	Y vs. E Men *p*-Value	E Men vs. Women *p*-Value
collagen content (% area) (10× magnification)	5.99 ± 1.34	10.0 2 ± 3.69	9.90 ± 1.50	0.001 ***	0.927
collagen content (% area) (20× magnification)	2.95 ± 1.59	6.87 ± 1.79	6.58 ± 1.12	0.001 ***	0.995
Fat infiltration	0	9 (75%)	12 (75%)	NA	NA
Muscle atrophy	0	10 (83%)	13 (81.25%)	NA	NA

Unpaired, two-sample *t*-test, NA: not applicable. Values are presented as numbers or means ± SD. Y: Young; E: Elderly; *** *p* ≤ 0.001.

**Table 3 ijms-23-11061-t003:** The percentage area of the elastic fibers and the AOD (average optical density) of COLI and COLIII in the skeletal muscle of the young and elderly male participants (n = 22).

Variable	Y Men (n = 10)	E Men (n = 12)	*p*-Value
Elastic fiber (% area)	7.99 ± 1.65	3.50 ± 1.66	0.001 ***
AOD of COLI	0.25 ± 0.04	0.37 ± 0.04	0.001 ***
AOD of COLIII	0.33 ± 0.09	0.28 ± 0.07	0.293

Unpaired, two-sample *t*-test, the values are presented as numbers or means ± SD, at 40× magnification. AOD: average optical density, COLI: collagen type I, COLIII: collagen type III, Y: Young; E: Elderly. *** *p* ≤ 0.001.

**Table 4 ijms-23-11061-t004:** The percentage area of collagen and the AOD (average optical density) of COLI and COLIII in the skeletal muscle of the mice.

Characteristic	Group A (n = 5)	Group B (n = 5)	Group C (n = 5)	A vs. B *p*-Value	A vs. C *p*-Value	B vs. C *p*-Value
^#^ Collagen content (% area)	2.05 ± 0.44	3.26 ± 0.86	7.97 ± 0.80	0.071	<0.001 ***	<0.001 ***
^#^ AOD of COLI	0.17 ± 0.01	0.22 ± 0.02	0.30 ± 0.03	0.013 *	0.002 **	<0.001 ***
^$^ AOD of COLIII	0.23 ± 0.08	0.23 ± 0.04	0.22 ± 0.05	0.994	0.897	0.982

ANOVA with Tukey post-hoc test ^#^/Games–Howell post-hoc test ^$^, AOD: average optical density, Group A = 6 weeks old (considered the pubertal group), Group B = 8 months old (considered the middle-aged group), Group C = 2 years old (considered the elderly group). The values are presented as numbers or means ± SD, at 40× magnification. * *p* < 0.05, ** *p* ≤ 0.01, *** *p* ≤ 0.001.

## Data Availability

Not applicable.

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
