# Peer review of "The Effects of Aging on the Intramuscular Connective Tissue"

_ijms, 2022, doi:10.3390/ijms231911061_

Round 1

Reviewer 1 Report

The clearly written manuscript “The Effect of Aging on the Intramuscular Connective Tissue” deals with an important aspect of aging. The study provides solid data of the effect of aging on the intramuscular ECM although the study is limited by the age and number of human samples. However, there are a few points that have to be clarified and to be improved before acceptance of the manuscript.

The authors estimated the collagen content between the different samples based on a picrosirius staining but it is necessary to use a biochemical approach like a total collagen assay in order to determine the collagen content. In general, the electron micrographs are not convincing as it is absolutely necessary to show the ECM and especially the collagen fibrils at a higher magnification. In the present state, the electron micrographs are useless. The differences in the distribution of collagen I and collagen III are important but what are the consequences at the ultrastructural level. Are there any changes in morphology, distribution and orientation of the collagen fibrils? There is an increase in collagen content but what does this means? Thicker collagen fibrils or more collagen fibrils? An immunogold labeling of individual collagen fibrils would give more precise information about the distribution of collagen I and collagen III. In this context, is it possible that the finding that there is no difference in collagen III is more based on a masking of epitopes for collagen III because of more collagen I on the surface of the collagen fibrils.

The authors stated in the discussion (line 203-213) that collagen III forms flexible meshworks. What is the evidence for that? Typically, collagen III is able to co-polymerize with collagen I to form heterotypic collagen fibrils. As already mentioned before, an immunogold labeling would help to show this at the ultrastructural level.

Author Response

The clearly written manuscript “The Effect of Aging on the Intramuscular Connective Tissue” deals with an important aspect of aging. The study provides solid data of the effect of aging on the intramuscular ECM although the study is limited by the age and number of human samples. However, there are a few points that have to be clarified and to be improved before acceptance of the manuscript.

The authors estimated the collagen content between the different samples based on a picrosirius staining but it is necessary to use a biochemical approach like a total collagen assay in order to determine the collagen content.

We thank the reviewer for this comment. It is true that we did not use any biochemical approach to quantify the total collagen, but we decided to use the Picrosirius red staining because we were interested to understand where the modifications of IMCT were localized, (in the endomysium and/or perimysium). By this approach it was very clear the increase of the collagen amount with aging (Figure 1 and Figure 6), permitting also to analyze the endomysium alone, as explained at Page 4: “The percentages of collagen fibers in the IMCT (endomysium plus perimysium) were: 5.99 ±1.34% in the young men, 10.02 ±3.69% in the elderly men, and 9.90 ±1.50% in the elderly women (Table 2). For the endomysium alone, the percentages were: 2.95 ±1.59% in the young men, 6.87 ±1.79% in the elderly men, and 6.58 ±1.12% in the elderly women. The collagen contents in the IMCT were significantly elevated in the elderly subjects (P=0.001) (Figure1 C-F, G; Figure 2; Table 2), with no significant differences according the gender (P>0.05) (Table 2).”

For this reason, we decided to go on with another approach for the estimation of the amount of the collagen types in the different samples: it was performed by the AOD (average optical density) after immunostaining with anti-COLI and anti-colIII, as explained in paragraph 4.8.

However, we agree with the reviewer that also a biochemical test will be very useful, so we added this point in the “Limitations and further research” paragraph in the Discussion, with this sentence: “it has not been explored the amount of HA compared to the other GAGs to better understand the relative modifications with aging, just as the collagen content in the different samples was not quantified with a biochemical approach.”

In general, the electron micrographs are not convincing as it is absolutely necessary to show the ECM and especially the collagen fibrils at a higher magnification. In the present state, the electron micrographs are useless. The differences in the distribution of collagen I and collagen III are important but what are the consequences at the ultrastructural level. Are there any changes in morphology, distribution and orientation of the collagen fibrils? There is an increase in collagen content but what does this means? Thicker collagen fibrils or more collagen fibrils? An immunogold labeling of individual collagen fibrils would give more precise information about the distribution of collagen I and collagen III. In this context, is it possible that the finding that there is no difference in collagen III is more based on a masking of epitopes for collagen III because of more collagen I on the surface of the collagen fibrils. The authors stated in the discussion (line 203-213) that collagen III forms flexible meshworks. What is the evidence for that? Typically, collagen III is able to co-polymerize with collagen I to form heterotypic collagen fibrils. As already mentioned before, an immunogold labeling would help to show this at the ultrastructural level.

Thank you for these very useful comments.

We have changed the TEM images (Figure 2) with more convincing ones at higher magnification: we hope that these new pictures can at least partially answer to the reviewer’s doubts. From the new pictures (Figure 2-C and 2-F) the bigger deposition and the higher density of collagen fibers in the IMCT of the elderly muscle sample are easily evident.

Certainly, it will be very interesting to make an analysis of the thickness of the fibers, but we have selected for this preliminary work only two samples of young men and two of elderly men, so we are not able now to give results statistically certain. From our preliminary images it seems that the fibers are not thicker, but they are much more dense with aging.

The masking of epitopes of collagen III and the formation of heterotypic fibrils could be also hypothesis that cannot be excluded by this work. However, it has already been demonstrated that with age and maturation there is a general and progressive shift in collagen type (toward more type I) in the muscle (McCormick RJ. The flexibility of the collagen compartment of muscle. Meat Sci. 1994;36(1-2):79-91), so we are led to think that a similar mechanism also occurs in IMCT. Moreover, we have added a new reference supporting our sentence affirming that collagen III forms flexible meshworks: in fact, Parkin and coauthors affirmed that “The collagen III fibril assumes a “flexi-rod” structure with flexible zones interspersed with rod-like domains, which is consistent with the molecule’s prominence in young, pliable tissues and distensible organs” (Parkin JD, San Antonio JD, Persikov AV, Dagher H, Dalgleish R, Jensen ST, Jeunemaitre X, Savige J. The collαgen III fibril has a "flexi-rod" structure of flexible sequences interspersed with rigid bioactive domains including two with hemostatic roles. PLoS One. 2017;12(7):e0175582).

Surely, the immunogold would be perfect to analyze better the samples and distinguish collagen type I from collagen type III: however, it would require additional time and costs, which are not feasible in the timing of the review of this paper. In the next future we have in program to perform a second harmonic generation analysis to visualize also the orientation, thickness and crimping of the collagen fibers: we have added this point on the paragraph 3.4 (Limitation and further research), together with the reviewer’s suggestion about the immunogold. The new sentence is: “Lastly, in this work we have no data about orientation, crimping and thickness of the collagen fibers at the ultrastructural level: for this reason, in the next future it would be necessary to perform a second harmonic generation analysis, together with an immu-nogold labeling of individual collagen fibers, to achieve more precise information about the distribution of collagen type I and III.”

Reviewer 2 Report

The authors should consider the following:

  The EM figures are difficult to resolve and could be improved. Better definition of collagen and elastic fibers in these figures is needed.

  It would be of interest to know the percentage of HA compared to the other GAGs ?

Author Response

The authors should consider the following:

The EM figures are difficult to resolve and could be improved. Better definition of collagen and elastic fibers in these figures is needed.

Authors agree with the reviewer and changed the Figure 3 (with better definition of the elastic fibers by TEM), and Figure 2, adding higher magnification pictures of collagen fibrils in young and elderly.

It would be of interest to know the percentage of HA compared to the other GAGs ?

We thank the reviewer for this good observation. It would actually be very useful and interesting to quantify the different GAGs and analyze the relative amount of HA. However, we know from our precedent works that the HA is the main GAG in the fascial tissue (Fede at al., 2021, Fede et al., 2018, Stecco et al., 2011, Stecco et al., 2013): for this reason we focused on it in this first work about aging. Therefore, the analysis of other GAGs is beyond the main objective of this work. Anyhow, we indirectly analyzed the quantity of hyaluronan using the Alcian blue stain with the CEC (critical electrolyte concentration) principle, in both young and elderly. As shown in Figure 7, A and B, the decreased staining associated with increased electrolyte concentration (MgCl2 from 0.05M in A, to 0.3 M in B) is due to competition of the cations of the salt with those of the dye for the polyanionic sites of the tissue: increasing the concentration of the salt the hyaluronan is no longer stained. This lends support the staining is due to HA-rich matrix, as explained in paragraph 2.3 and by Scott and Dorling (1965) (Reference 22 of the paper).

However, we added a sentence in paragraph 2.3 to better explicit this point; the new sentence is: “Alcian Blue staining showed that the IMCT was rich in glycosaminoglycans (Figure 7 A, Alcian Blue 0.05% with MgCl2 0.05 M). Furthermore, the decreased staining associated with increased electrolyte concentration (Figure 7 B, Alcian Blue 0.05% with MgCl2 0.3 M) lends support to the staining is due to HA-rich matrix, being the HA selectively stained only at low concentration of salt (data obtained both in young and elderly, not shown) [22].”

We added also the possibility to further investigate the other GAGs in the “Limitations and further research” paragraph in the Discussion: “Thirdly, it has not been explored the amount of HA compared to the other GAGs to better understand the relative modifications with aging.”

Round 2

Reviewer 1 Report

The manuscript is ready for publication although I still do not believe that the proposed collagen III containing flexible meshworks really exists. One have to show this by electron microscopy in combination with immunogold labeling. There is no publication available showing this at high resolutution. You are right that Parkin et al. have shown that collagen III has a flexi-rod structure (molecule structure) but it is not shown in this publication that meshwork-like structures are formed by collagen III. I am not sure that second-harmonic generation microscopy will give you the necessary information about the suprastructural assembly.